# Vertical and Lateral Etch Survey of Ferroelectric AlN/Al_1−x_Sc_x_N in Aqueous KOH Solutions

**DOI:** 10.3390/mi13071066

**Published:** 2022-07-02

**Authors:** Zichen Tang, Giovanni Esteves, Jeffrey Zheng, Roy H. Olsson

**Affiliations:** 1Electrical and Systems Engineering Department, University of Pennsylvania, Philadelphia, PA 19104, USA; zichent@seas.upenn.edu; 2Microsystems Engineering, Science and Applications (MESA), Sandia National Laboratories, Albuquerque, NM 87123, USA; gesteve@sandia.gov; 3Material Science and Engineering Department, University of Pennsylvania, Philadelphia, PA 19104, USA; jxzheng@seas.upenn.edu

**Keywords:** aluminum scandium nitride (AlScN), aluminum nitride (AlN), wet etch, potassium hydroxide (KOH), ferroelectric

## Abstract

Due to their favorable electromechanical properties, such as high sound velocity, low dielectric permittivity and high electromechanical coupling, Aluminum Nitride (AlN) and Aluminum Scandium Nitride (Al_1−x_Sc_x_N) thin films have achieved widespread application in radio frequency (RF) acoustic devices. The resistance to etching at high scandium alloying, however, has inhibited the realization of devices able to exploit the highest electromechanical coupling coefficients. In this work, we investigated the vertical and lateral etch rates of sputtered AlN and Al_1−x_Sc_x_N with Sc concentration x ranging from 0 to 0.42 in aqueous potassium hydroxide (KOH). Etch rates and the sidewall angles were reported at different temperatures and KOH concentrations. We found that the trends of the etch rate were unanimous: while the vertical etch rate decreases with increasing Sc alloying, the lateral etch rate exhibits a V-shaped transition with a minimum etch rate at x = 0.125. By performing an etch on an 800 nm thick Al_0.875_Sc_0.125_N film with 10 wt% KOH at 65 °C for 20 min, a vertical sidewall was formed by exploiting the ratio of the $\left\{10\bar{11}\right\}$ planes and $\left\{1\bar{1}00\right\}$ planes etch rates. This method does not require preliminary processing and is potentially beneficial for the fabrication of lamb wave resonators (LWRs) or other microelectromechanical systems (MEMS) structures, laser mirrors and Ultraviolet Light-Emitting Diodes (UV-LEDs). It was demonstrated that the sidewall angle tracks the trajectory that follows the $\left\{\bar{1}2\bar{12}\right\}$ of the hexagonal crystal structure when different *c*/*a* ratios were considered for elevated Sc alloying levels, which may be used as a convenient tool for structure/composition analysis.

## 1. Introduction

Acoustic filters are key components in the evolution of radio frequency communication systems. Quartz crystals, lithium tantalate (LiTiO_3_) and lithium niobate (LiNbO_3_) surface acoustic wave (SAW) filters made the first two generations (Global System for Mobile communication, or GSM and Code-Division Multiple Access, or CDMA) of mobile networks possible [1,2], followed by AlN Bulk Acoustic Wave (BAW) resonators for the 3G (WCDMA) and 4G-LTE networks [3]. As the road map to 5G and beyond unfolded, filter requirements have increased and call for filters that exhibit lower insertion loss, higher temperature stability, steeper skirts and wider bandwidth. To achieve these metrics, a new material with enhanced piezoelectric coefficients is needed, since it can lead to more efficient coupling that directly transfers to increased filter bandwidth. In 2001, Takeuchi identified, by first principal calculation, that wurtzite AlScN had the potential [4] to achieve higher piezoelectric coefficients than AlN. Akiyama et al. later showed this through measurement of co-sputtered AlScN films [5,6,7] that demonstrated a peak d_33_ of 27.6 pC/N [6]. Therefore, AlScN has become a competitive candidate for different filter designs targeting 5G NR mmWave (K_a_ band, 26 GHz) [8,9,10] or higher. As a result, extensive research has been conducted on its material properties, growth, characterization and device fabrication.

Etching is a key step in the fabrication of AlScN devices. Like other III–V nitride alloys, existing etching techniques on AlN/Al_1−x_Sc_x_N can be grouped into two categories: dry etching and wet etching. For dry etching, ion-milling that uses Ar exclusively and Inductively Coupled Plasma Reactive Ion Etching (ICP-RIE), which utilizes BCl_3_/Cl_2_/Ar mixtures, are common methods [11]. The latter is the more routinely used technique for dry etching polycrystalline AlN, and etch rates can reach up to 420 nm/min [12] for an ICP power of 800 W. Nevertheless, this etch rate drops dramatically with increasing scandium concentration. For an Al_0.85_Sc_0.15_N film, the etch rate declined to 64% of that of AlN [13]; for Al_0.73_Sc_0.27_N, 42% under the same etch condition [14]; and for Al_0.64_Sc_0.36_N, 10% of that for AlN [15]. As for single-crystalline Al_1−x_Sc_x_N, the reduction in etch rate occurred much faster: at x = 0.02, the etch rate already reduced to 15% of AlN, and at x = 0.15, it was 12.7% [16]. Not only has the existence of scandium retarded the etch rate, but its non-volatile etching by-products also re-deposit during the etch process, resulting in a roughened and tapered side wall less than 76° if Ion Beam Etching (IBE) is not used [17,18]. The poor selectivity requires very thick, hard masks during processing and makes it challenging to stop the AlScN etch on underlying metal electrode materials. Both the slow etch rate and low selectivity can limit the maximum AlScN film thickness realizable in a MEMS process. Table 1 summarizes selected studies on the dry etch rates of AlN/Al_1−x_Sc_x_N:

Compared to dry etching, wet etching can be rather advantageous owing to its less expensive tooling. Common etchants include tetramethylammonium hydroxide (TMAH) [19,20,21], phosphoric acid (H_3_PO_4_) [22,23] or phosphoric acid-based solution [21] (PWS, containing 80% H_3_PO_4_, 16% H_2_O and 4% HNO_3_), potassium hydroxide (KOH) [22,24,25] and AZ400K Developer (contain 10 wt% of KOH) [26]. Among them, we found KOH to be rather attractive due to its availability and non-toxicity with a relatively high etch rate and high etch selectivity to silicon nitride (SiN_x_). The current research focusing on AlN etching in KOH is abundant, yet very few details were disclosed regarding the etching of Al_1−x_Sc_x_N, let alone a complete survey of its etch rate as a function of Sc alloying. For the data available, the etch rate varies substantially based on the temperature, crystallinity and etchant concentration. Table 2 summarizes previous studies on the etch rates of AlN/Al_1−x_Sc_x_N in aqueous KOH.

The lack of variable control and the limited data make it difficult to draw solid conclusions on the factors affecting the etch rate, though from the AlN/Al_1−x_Sc_x_N dry etch results and the KOH wet etch of its III–V alloy kin Al_1−x_In_x_N [27]/Al_1−x_Ga_x_N [29], one would assume that the etch rate decreases with the increasing Sc concentration. Moreover, even though prior etching studies reported the anisotropic nature of the etch, with preferential etching of the c-plane $\left\{000\bar{1}\right\}\;$ in N-polar AlN/Al_1−x_Sc_x_N [28,29] while exposing the $\left\{10\bar{11}\right\}\;$ planes [31] from whose boundary the sidewall that follows is the $\left\{\bar{1}2\bar{12}\right\}\;$ of the hexagonal crystal structure, lateral etch rates have not been reported. To date, there have been but a few papers [32,33] discussing KOH etching of AlN/Al_1−x_Sc_x_N in detail, and only AlN and Al_0.80_Sc_0.20_N were studied. Therefore, we report a thorough survey on the vertical and lateral etch rates of sputtered AlN/Al_1−x_Sc_x_N in aqueous KOH solutions vs. scandium concentrations, where the KOH concentrations and solution temperatures were strictly regulated. We found that the vertical etch rate declines steadily with increasing scandium concentration, whereas the lateral etch rate experiences a V-shaped transition with a minimum value of 0.043 ± 0.002 nm/s at x = 0.125. By etching the Al_0.875_Sc_0.125_N film in 10 wt% KOH at 45 °C for 10 min, a nearly 90° sidewall was produced by exposing the $\left\{1\bar{1}00\right\}$ planes. This technique is capable of generating a vertical sidewall without pre-treatment and could be beneficial for the fabrication of numerous kinds of microelectromechanical systems (MEMS) device such as lamb wave resonators (LWRs), laser mirrors and Ultraviolet Light-Emitting Diodes (UV-LEDs).

The findings are presented in four sections: (1) Introduction, containing the problem selected, literature review, novelty and section description; (2) Experiment, containing the deposition methods, film characterization, etch mask patterning, wet etching process steps and data interpretation methodology; (3) Results and Analysis, containing the illustration of results and detailed analysis; (4) Conclusion, summarizes the paper and the novelty of the research.

## 2. Experiment

### 2.1. Film Deposition

#### 2.1.1. Growth

Based on the crystallinity of AlN/Al_1−x_Sc_x_N, several methods can be used for film growth. Single-crystal AlN can be grown via physical vapor transport (PVT) on substrates up to 60 mm in diameter [34,35,36]. Single-crystal Al_1−x_Sc_x_N can be grown by molecular beam epitaxy (MBE) [9] on 100 mm wafers [37] with scandium concentration x from 0.06 to 0.36 [38]. The growth of Sc alloyed AlN via metal–organic chemical vapor deposition (MOCVD) had suffered from the lack of Sc precursors [39]; however, research in this field is catching up quickly, and the ability to grow 36% Sc alloyed Al_1−x_Sc_x_N films on 100 mm wafers has been demonstrated [40,41]. High-quality polycrystalline Al_1−x_Sc_x_N films can be deposited with physical vapor deposition (PVD) methods such as magnetron sputtering. This method features a high deposition rate, low growth temperature, and the capability of up-scaling in substrate size [42,43,44,45,46], and has been adopted by a variety of tool manufacturers for industrial mass production [47]. 

Al_1−x_Sc_x_N depositions were performed in an Evatec CLUSTERLINE^®^ 200 II Physical Vapor Deposition System (Evatec AG, Trübbach, Switzerland) at a substrate temperature of 350 °C with 150 kHz pulsed DC bias with an 88% duty cycle. No RF substrate bias was applied and the AlScN materials were deposited directly onto silicon substrates. No pre-cleans were applied to the substrates. Two 100 mm metal targets were used, Al (99.999%) and Sc (99.99%), with a target-to-substrate distance of 88.5 mm. Before deposition, the chamber was pumped to a base pressure lower than 1.0 × 10^−7^ mbar. A 15 nm AlN film was deposited first onto a 100 mm Si (100) wafer as the seed layer by sputtering Al in a pure nitrogen environment with a target power of 1000 W and N_2_ flow of 20 sccm. Subsequently, a 35 nm linearly graded Al_1−x_Sc_x_N layer was deposited by gradually increasing the Sc target power while maintaining the Al target power constant. Finally, a 750 nm bulk Al_1−x_Sc_x_N layer was deposited by fixing both the Al and Sc target powers. The 20 sccm N_2_ flow was maintained during the process and no Ar was used throughout. The chamber pressure remained close to 8.0 × 10^−4^ mbar during the deposition. A total of 15 films were deposited with Sc concentration ranging from 0% to 42%. Table 3 summarizes the correlation between Sc target power and Sc concentration based on our previous research [48].

#### 2.1.2. Surface Metrology

The adoption of the AlN seed layer, the Al_1−x_Sc_x_N gradient layer, and pure nitrogen sputtering environment greatly reduced the occurrence of abnormally oriented grains (AOGs). AOGs are a series of wurtzite Al_1−x_Sc_x_N crystals that do not have their c-axis perpendicular to the substrate [49]. They could erupt from the crystalline interface if grown under unfavorable conditions, especially for films with higher Sc concentration [50]. If not suppressed, they may occupy the entire film surface [49,51,52], severely degrading device performance [53,54] and locally slowing the etch rate. To examine the film quality, atomic force microscopy (AFM) scans were conducted using a Bruker Icon AFM, and most of the films measured showed a roughness of <2 nm. Figure 1a shows the surface of an Al_0.64_Sc_0.36_N film within a 20 × 20 μm^2^ field. The root mean square (RMS) surface roughness R_q_ is 0.840 nm and R_a_ is 0.641 nm. The film quality is also supported by the rocking curve measurements, which were performed by a Rigaku Smart Lab X-RAY Diffractometer (XRD) with a high resolution Parallel Beam (PB) Ge (220) × 2 monochromator (Rigaku Corporation, Tokyo, Japan). The Omega scan data are centered at 18.13 ° with a full width at half maximum (FWHM) of 1.80 ° for a 500 nm film deposited on Si (100) using the same process, indicating that the film is highly *c*-axis textured. All films had a similar quality based on the AFM and XRD measurement. 

### 2.2. Film Patterning

To selectively etch AlN/Al_1−x_Sc_x_N, the film must be patterned so that only locations of interest would be exposed in the etchant. Silicon nitride (SiN_x_) was chosen as our hard mask, as it has an etch rate of 0.67 nm/min in 30% KOH at 80 °C [55], which is negligible compared to the etch rates of Al_1−x_Sc_x_N. A 200 nm SiN_x_ film was deposited on top of the Al_1−x_Sc_x_N film in an Oxford Plasma Lab 100 Plasma-enhanced Chemical Vapor Deposition (PECVD) machine (Oxford Instruments Plasma Technology, Bristol, UK), and on the backside of the wafer as well to protect the Si substrate during the etch. Afterwards, photoresist MICROPOSIT^®^S1813 was spin-coated and exposed in a Karl Süss MA6 Mask Aligner (SÜSS MicroTec SE, Garching, Germany) via contact lithography. The exposed film was developed in TMAH-0.26N developer, then transferred to an Oxford 80 plus Reactive-ion Etching (RIE) etcher (Oxford Instruments Plasma Technology, Bristol, UK). A 30 s O_2_ descum was performed first to remove the remaining resist, followed by 3 min of SiN_x_ etch with CHF_3_/O_2_ mixture. Finally, the resist was stripped in MICROPOSIT^®^ remover 1165 with ultrasonic bath at 60 °C and plasma cleaned. Figure 2 illustrates the fabrication process.

### 2.3. KOH Wet Etching

#### 2.3.1. Principle

The wet etching of III–V nitrides in general involves the formation of an oxide on the surface and the subsequent dissolution of the oxide [24]. The flowing reactions occur when AlN/Al_1−x_Sc_x_N is subjected to the alkaline environment [29]:(1)$$2\mathrm{AlN}+3H_{2}O\overset{\;\;\;\;\mathrm{KOH}\;\;\;\;}{\to }Al_{2}O_{3}+2NH_{3}$$
(2)$$\mathrm{AlN}+3H_{2}O\overset{\;\;\;\;\mathrm{KOH}\;\;\;\;}{\to }\mathrm{Al}{\left(\mathrm{OH}\right)}_{3}+NH_{3}$$
(3)$$2ScN+3H_{2}O\overset{\;\;\;\;\mathrm{KOH}\;\;\;\;}{\to }Sc_{2}O_{3}+2NH_{3}$$
(4)$$ScN+3H_{2}O\overset{\;\;\;\;KOH\;\;\;\;}{\to }Sc{\left(OH\right)}_{3}+NH_{3}$$

In this reaction, KOH acts as the catalyst that pushes the equation to the right side. Due to the origin of the reaction, N-polar AlN/Al_1−x_Sc_x_N are preferred to be etched as it is difficult for $OH^{-}$ to make contact with the Al/Sc atoms in the metal polar state because of repulsion from the negatively charged dangling nitrogen bonds [24].

#### 2.3.2. Etching Process

A water bath was established for the etching to be conducted in a stable temperature environment. A trough was filled with deionized (DI) water and placed upon an Echo Thermal HP30 hotplate (Torrey Pines Scientific, Inc., Carlsbad, California). Around 300 mL of 30% KOH/diluted KOH was poured into a beaker, which was later transferred into the trough. A Teflon plate was added in between to avoid direct heating. Finally, a thermocouple was submerged into the KOH solution and connected to the hotplate through a proportional–integral–derivative (PID) control loop to adjust the solution temperature. With all these measures, the solution temperature was able to be stabilized within ±1 °C during etching. The test sample was cleaved into an 8 × 10 mm^2^ die and clamped by a tweezer when dipped into the solution. When submerged in the KOH, samples experienced minimal agitation, and upon removal, they were rinsed under DI water and subsequently dried with N_2_. Finally, the sample was cleaved from the middle (Figure 3a) and a cross-section was imaged in a FEI Quanta 600 Environmental Scanning Electron Microscope (ESEM) (FEI Company, Hillsboro, OR, USA) (Figure 3b). 

#### 2.3.3. Data Interpretation

Three types of data were extracted from the SEM images: vertical etch depth, lateral etch length (undercutting) and the sidewall angle. The vertical etch depth is defined as the etching depth into the AlN/Al_1−x_Sc_x_N film from the bottom of the SiN_x_ hard mask. It can be somewhat ambiguous, as hexagonal-shaped hillocks are known to form after KOH etching [29], which makes it difficult to identify the end point of the etch. Since the phenomenon is presumed to be defect-related in AlN [28,55], we conclude that the ‘tip’ of the pyramid acts as a mask in this process and thus the etch front should be read as the basal plane of the hexagonal pyramid. Figure 4a is a demonstration of how the etch depth was measured in case of the existence of the hillocks. The lateral etch length is defined as the etch length underneath the SiN_x_ hard mask (Figure 4b). Finally, the sidewall angle is simply the angle between the sidewall and base plane. 

## 3. Results and Analysis

### 3.1. Etch Result with 30% KOH at 45 °C

Due to the limited film thickness and the vastly different etch rates of films with varying Sc concentrations, it was not practical to use the same etch time when determining the vertical etch rate. Instead, films with lower Sc concentration were etched for a shorter amount of time. The following table (Table 4) illustrates the time used in each case:

It should be noted that not all samples fabricated, as mentioned in Section 2, had their vertical etch rate measured due to the availability of the sample at the time this experiment was conducted. For the etched samples, cross-section images were taken from several spots to avoid local variations. Figure 5 summarizes the vertical and lateral etch rates vs. Sc concentration for etching in 30% KOH at 45 °C.

Intuitively, the vertical etch rate matches our expectation: a steady decline with increasing Sc concentration. The Al_0.80_Sc_0.20_N has an average vertical etch rate of 7.58 nm/s, and the Al_0.64_Sc_0.36_N has an average vertical etch rate of 3.68 nm/s. Compared to the values of 3.59 nm/s and 2.77 nm/s obtained by K. Bespalova et al. [25] and S. Fichtner et al. [23] respectively, they are not exact matches but in the same order of magnitude. It should be noted that as per the view of A. Ababneh et al. [56], the etch rate of AlN is a strong function of sputtering conditions; thus, this study can be used to predict the trend instead of the absolute etch rate when applied to films deposited under different conditions. The lateral etch rate has several distinctive features requiring further examination. First, the standard deviation of the lateral etch rate is considerably larger than that for the vertical etch. This is due to the finite amount of lateral etching performed. As exhibited in Figure 4a, the film has a lateral etch length of only a few dozen nanometers in some cases, which makes it difficult to accurately measure. Secondly, the etch rate of Al_0.72_Sc_0.28_N does not fit in the line. Lastly, the lateral etch rate begins to increase for concentrations in excess of 20% Sc.

To further explore the lateral etching, a second experiment was conducted in which the lateral etch length was extracted over a longer etching period. The number of Sc concentrations in this study was also expanded for higher resolution as shown in Table 5: 

Before the long-span etching was performed, a linearity check was conducted to make sure that the etch does not change during the etching process. The film tested was Al_0.64_Sc_0.36_N and it was subjected to etches of 60 s, 80 s, 300 s, 1200 s and 2400 s, respectively. A linear regression was performed, and the lateral etching proved to be highly linear with the R^2^ = 0.9968. The fact that the same etch rate was retained after 40 min indicates that the etch is reaction-limited, e.g., not confined by mass transferring, and matches the description of AlN etching in KOH given by Mileham et al. [26] (Figure 6).

The long etch time returned similar results compared with the short etch time. The etch rate experienced a transition where it reaches the lowest at x = 0.125. However, discrepancies still exist for the etch rates of x = 0.28, 0.30, 0.34 and 0.38 when compared to other films with Sc concentration x > 0.125. One explanation might be that these films were deposited in different batches as opposed to the rest of the films. As stated above, what exhibits as different etch rates here might be the result of the different chamber condition when the films were being sputtered. It was apparent that even though they do not fit in the line made from the x = 0.125, 0.15, 0.20, 0.25, 0.32 and 0.36 films, they were able to constitute their own trend line with almost has the same slope. Deeper examination revealed that the etch rate of the x = 0.28 film was lower than expected on two different films deposited in different batches, and further studies should be carried out to better understand these subtle trends in lateral etch rate. 

For most Sc concentrations, the sidewall angle of films remains invariant throughout the etching. As demonstrated in Figure 7, an extra-long submerge of the sample does not change the sidewall angle by a visible amount, and for the changes that could be measured, it can be attributed to the tilting of the sample itself during imaging.

The sidewall is a reflection of the crystal structure of the Al_1−x_Sc_x_N films. As per the findings of W. Guo et al., due to the energy difference between crystalline planes, the c-plane $\;\left\{000\bar{1}\right\}$ will be etched first prior to the deterioration of the $\left\{10\bar{11}\right\}$ planes [31]. Hence, during the anisotropic etching process, the exposed 〈$\bar{11}23$〉 slip edges between the boundary of the $\left\{1\bar{011}\right\}$ planes forms the facets that follow the $\left\{\bar{1}2\bar{12}\right\}$ of the hexagonal crystal structure, behind which lateral etching ceases advancing (Figure 8).

Moreover, for a Hexagonal Close-Packed (HCP) unit cell with an axis length *c*/*a*, the sidewall angle, *θ* (Figure 9), of the $\left\{\bar{1}2\bar{12}\right\}$ facets as a function of lattice length can be calculated as $\theta =\arctan\left(c/a\right)$:

Using the *c*/*a* data from the work presented by Österlund et al. [45], and considering the isotropic etching of the $\left\{10\bar{11}\right\}$ plane, the theoretical sidewall angle *φ* can be plotted against the experimental value (Figure 10).

The absolute value of the experimental and theoretical angles follows the same trajectory with a parabolic downtrend with increasing Sc concentration until the Sc concentration exceeds 40%. We hypothesize that the side profile is the result of both anisotropic and isotropic etching. At lower Sc concentration, the etch rate on the c-plane $\left\{000\bar{1}\right\}$ is significantly higher, the momentum of the etch is downwards, and thus the side wall creates a facet that follows the $\left\{\bar{1}2\bar{12}\right\}$ of the hexagonal crystal structure. At a higher Sc concentration, the low vertical etch rate slows down the descending penetration, allowing the etchant to further react with the sidewall planes already exposed. Therefore, the closer the $\left\{10\bar{11}\right\}$ planes are to the surface, the more they are etched away; as a result, the sidewall angle becomes lower than that predicted solely based on the anisotropic crystal etching, i.e., lower than *θ*. This can be partially verified by some of the abnormal points on the graph, most of which have a small lateral etch rate (e.g., 12.5% and 42% Sc), which prevents the etching of their corresponding $\left\{10\bar{11}\right\}$ planes.

### 3.2. Etch Results with 30 wt% KOH at 65 °C and 10 wt% KOH at 65 °C

The experiment was also carried out at an elevated temperature and with lower KOH concentration to rule out any possible interference to the outcome except for the intrinsic material properties. At elevated temperature, the etching was performed with a short etching time of 150 s, except for Al_0.875_Sc_0.125_N, which was etched for 450 s. Etching with lower KOH concentration was performed for 20 min for all Sc alloying concentrations. As a result of the faster etching rate, the vertical etch rate could not be measured. The lateral etch rate and sidewall angle of the experiment are presented below (Figure 11): 

A trend consistent with the etching studies reported above was also observed here, where the lowest etch rate was found to be at x = 0.125 and a decreasing sidewall angle was observed with increasing Sc concentration. Under the SEM, white clusters can be seen occasionally near the etch front, which we assume to be the unsoluable reaction by-product Sc(OH)_3_. Further research still needs to be conducted on the effect of its presence. 

### 3.3. Formation of Vertical Sidewall in Al_0.875_Sc_0.125_N

While differences can be observed across the spectrum, the lateral etch rate unanimously reaches its lowest point when the Sc concentration is at 12.5%. An inspection of the etch results have shown that during the etching, a more vertical side wall can be formed, as shown in Figure 12.

Although (a) was etched with different parameters compared to (b–d)—plus, (b–d) were etched for almost the same time—we conjecture that these images are demonstrating the transient response of the same etching dynamics. The KOH etch is slowed significantly at the $\mathrm{Si}\;\left\{111\right\}$ plane and the Al_0.875_Sc_0.125_N $\left\{10\bar{11}\right\}$ plane due to the low etch rate. Because of the energy differences between removing $\left\{10\bar{11}\right\}$ Al_0.875_Sc_0.125_N and $\left\{111\right\}\;\mathrm{Si}$, the KOH slowly etches $\mathrm{Si}\;\left\{111\right\}$ until it comes into contact with the Al_0.875_Sc_0.125_N $\left\{1\bar{1}00\right\}$ planes, which requires lower energy to react than the $\left\{10\bar{11}\right\}$ planes. As a result, the etchant begins to remove and simultaneously etch $\mathrm{Si}\;\left\{111\right\}$ and $\left\{1\bar{1}00\right\}$ AlScN. By the time the entire $\left\{1\bar{1}00\right\}$ planes were exposed, a vertical sidewall was formed. It is possible that the slow lateral etch rate is necessary but not sufficient for the exposure of the $\left\{1\bar{1}00\right\}$ AlScN to occur. This has been demonstrated with the lateral etching of Al_0.85_Sc_0.15_N and Al_0.72_Sc_0.28_N, which have slightly higher lateral etch rates than Al_0.875_Sc_0.125_N. As shown below (Figure 13), the etchant may preferentially etch Si $\left\{111\right\}$ instead of m-plane Al_1−x_Sc_x_N. 

The $\left\{1\bar{1}00\right\}\;$ plane etching has been reported before in single-crystal GaN [57] and Al_1−x_Ga_x_N [58], and as per the findings of W. Chen et al. [57], the preference of its etching in GaN is a result of its smaller dangling bond density, which makes it more stable in KOH than the $\left\{10\bar{11}\right\}$ plane. Therefore, one explanation might be that the Al_0.875_Sc_0.125_N film has the highest activation energy in terms of the lateral etching, which was calculated to be 23.14 kcal/mol based on the data available. Nevertheless, this is one of the limited examples where this is reported in sputtered Al_1−x_Sc_x_N, and more research needs to be conducted to reveal the mechanism behind the vertical sidewall formation. This method, combined with BCl_3_/Cl_2_ dry etching, could potentially be applied in fabricating vertical side walls using a two-step fabrication process, which will benefit the research and production of LWR, laser mirrors, UV LEDs and a variety of MEMS devices.

## 4. Conclusions

We extensively studied the vertical and lateral etch rate of AlN/Al_1−x_Sc_x_N in aqueous KOH solutions across etch temperature, KOH concentration and a broad range of scandium alloying. It was shown that the vertical etch rate declines steadily with increased levels of Sc alloying, declining from 124.6 ± 0.68 nm/s for AlN to 3.7 ± 0.063 nm/s for Al_0.64_Sc_0.36_N in 30 wt% KOH at 45 °C. By contrast, the resistance to lateral etching peaks at a mere 0.043 ± 0.002 nm/s when x = 0.125. This is orders of magnitude lower compared to the lateral etch rate of 1.99 ± 0.01 nm/s for AlN or 1.99 ± 0.17 nm/s for Al_0.64_Sc_0.36_N. We have also demonstrated that KOH wet etching of Al_1−x_Sc_x_N is mostly anisotropic, and that the etch profile can be predicted from the crystal structure coupled with a small-scale isotropic etching of the sidewall. A technique for fabricating a vertical sidewall by exposing the $\left\{1\bar{1}00\right\}\;$ planes of sputtered Al_1−x_Sc_x_N was also demonstrated via etching an 800 nm thick Al_0.875_Sc_0.125_N film in 10 wt% KOH at 65 °C for 20 min. With this method, the fabrication of numerous MEMS devices such as LWRs, laser mirrors and UV-LEDs can be benefited. Future work will include detailed research on the activation energy for the lateral etching of AlN/Al_1−x_Sc_x_N using Arrhenius plots formed [32] from a series of design of experiments (DOE) using the Taguchi method [59].

## Figures and Tables

**Figure 1 micromachines-13-01066-f001:**
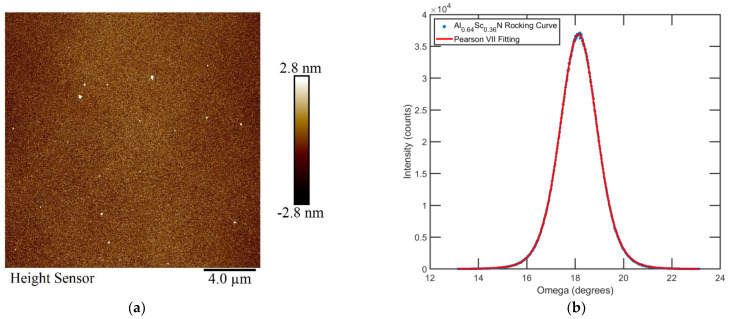
(**a**) AFM of the Al_0.64_Sc_0.36_N Film. (**b**) Rocking curve measurement (Omega Scan) of an Al_0.64_Sc_0.36_N film.

**Figure 2 micromachines-13-01066-f002:**
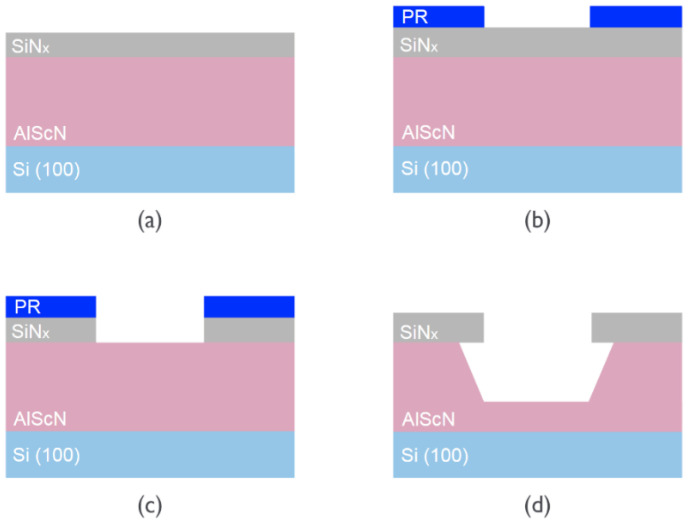
(**a**) Al_1−x_Sc_x_N and SiN_x_ hard mask deposition; (**b**) photoresist patterning; (**c**) SiN_x_ dry etch; (**d**) resist stripping and Al_1−x_Sc_x_N KOH wet etch. The opening area for the etching is 30 × 200 µm^2^.

**Figure 3 micromachines-13-01066-f003:**
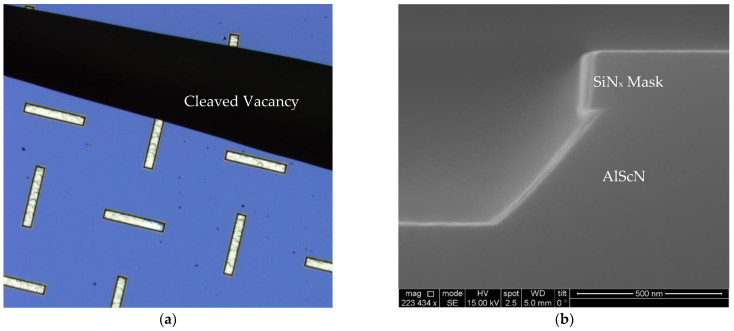
(**a**) A cleaved sample after etching. Black background is the vacancy after cleaving. (**b**) Cross-section SEM image of an Al_0.72_Sc_0.28_N film etched for 30 s in 30% KOH at 60 °C.

**Figure 4 micromachines-13-01066-f004:**
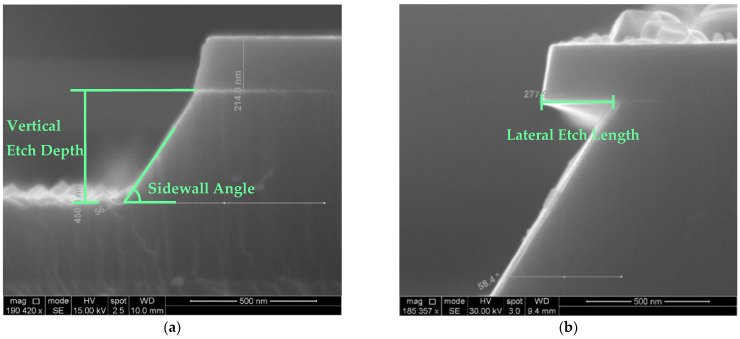
(**a**) Al_0.95_Sc_0.05_N etched for 10 s in 30% KOH at 45 °C. (**b**) Al_0.85_Sc_0.15_N etched for 2.5 min in 30% KOH at 65 °C.

**Figure 5 micromachines-13-01066-f005:**
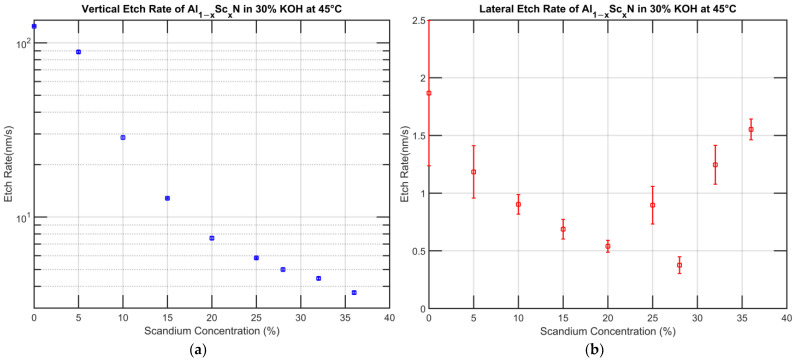
(**a**). Vertical etch rate of Al_1−x_Sc_x_N in 30% KOH at 45 °C. (**b**) Lateral etch rate of Al_1−x_Sc_x_N in 30% KOH at 45 °C (short etch time).

**Figure 6 micromachines-13-01066-f006:**
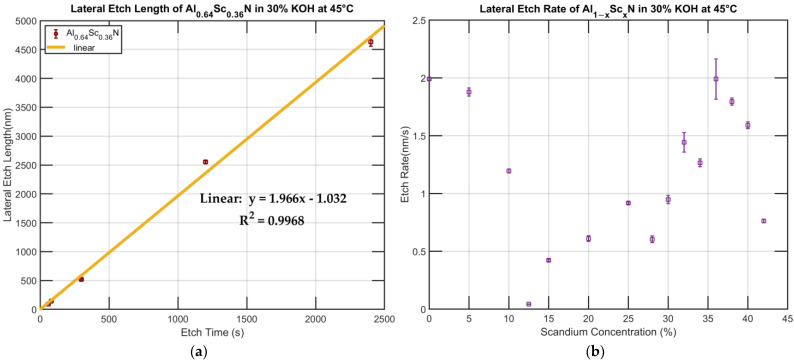
(**a**). Lateral etch length with respect to etch time of Al_0.64_Sc_0.36_N in 30% KOH at 45 °C. (**b**) Lateral etch rate of Al_1−x_Sc_x_N in 30% KOH at 45 °C (long etch time).

**Figure 7 micromachines-13-01066-f007:**
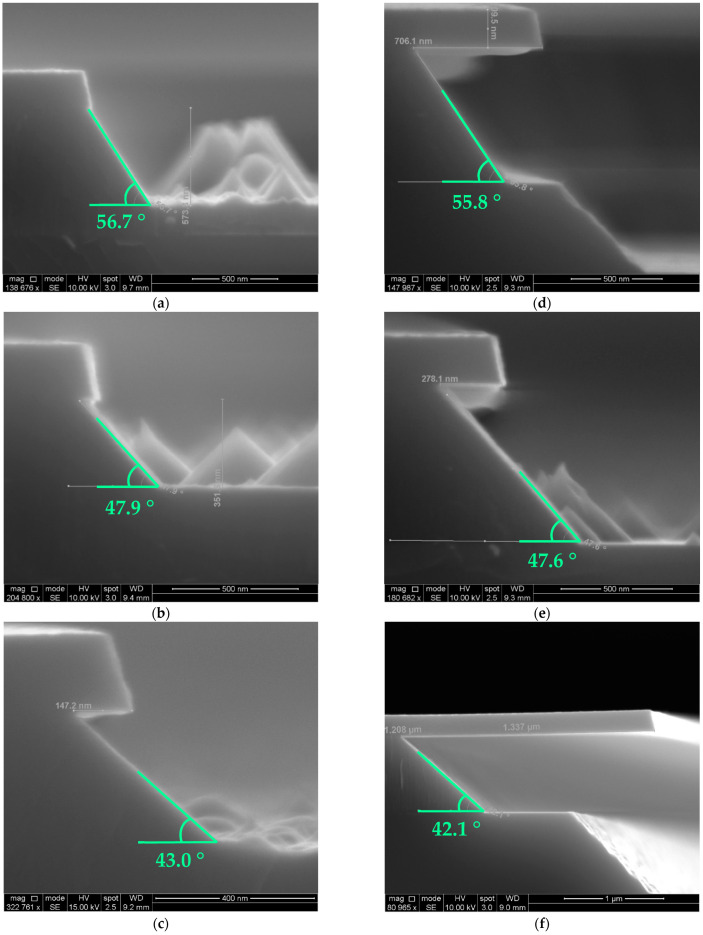
(**a**) Al_0.90_Sc_0.10_N etched for 20 s; (**b**) Al_0.75_Sc_0.25_N etched for 60 s; (**c**) Al_0.64_Sc_0.36_N etched for 80 s; (**d**) Al_0.90_Sc_0.10_N etched for 10 min; (**e**) Al_0.75_Sc_0.25_N etched for 5 min; (**f**) Al_0.64_Sc_0.36_N etched for 20 min. All etching was performed in 30 wt% KOH at 45 °C. The sidewall angle is preserved after the long etch.

**Figure 8 micromachines-13-01066-f008:**
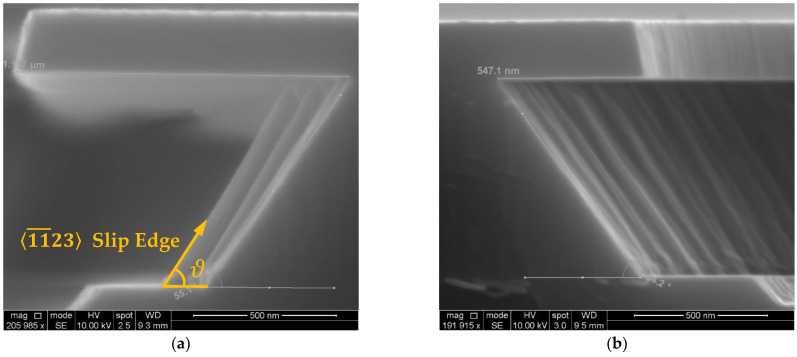
(**a**). Exposed 〈$\bar{11}23$〉 slip edge in an Al_0.95_Sc_0.05_N film etched in 30% KOH at 45 °C for 10 min; (**b**) the sidewall that follows the $\left\{\bar{1}2\bar{12}\right\}$ facet created by the boundary between the $\left\{10\bar{11}\right\}$ planes of the same etch condition for 5 min.

**Figure 9 micromachines-13-01066-f009:**
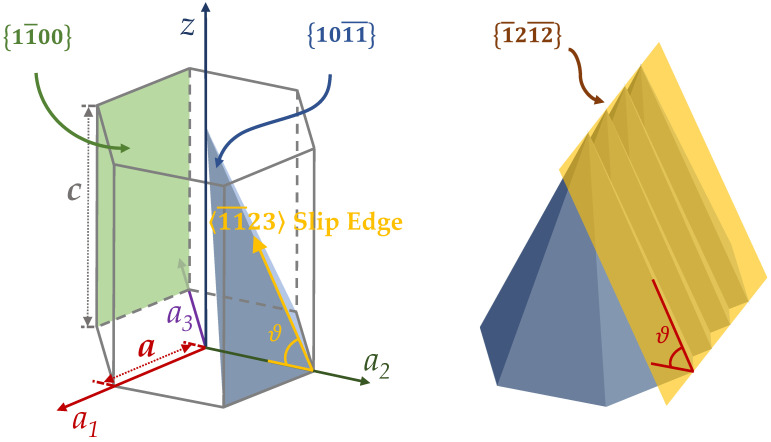
Crystal planes in Al_1−x_Sc_x_N HCP lattice.

**Figure 10 micromachines-13-01066-f010:**
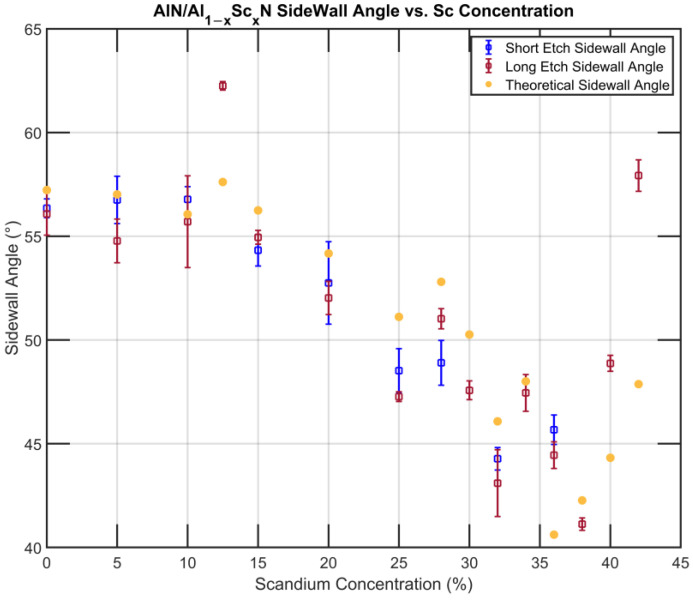
Comparison of experimental and theoretical sidewall angle vs. Sc concentration.

**Figure 11 micromachines-13-01066-f011:**
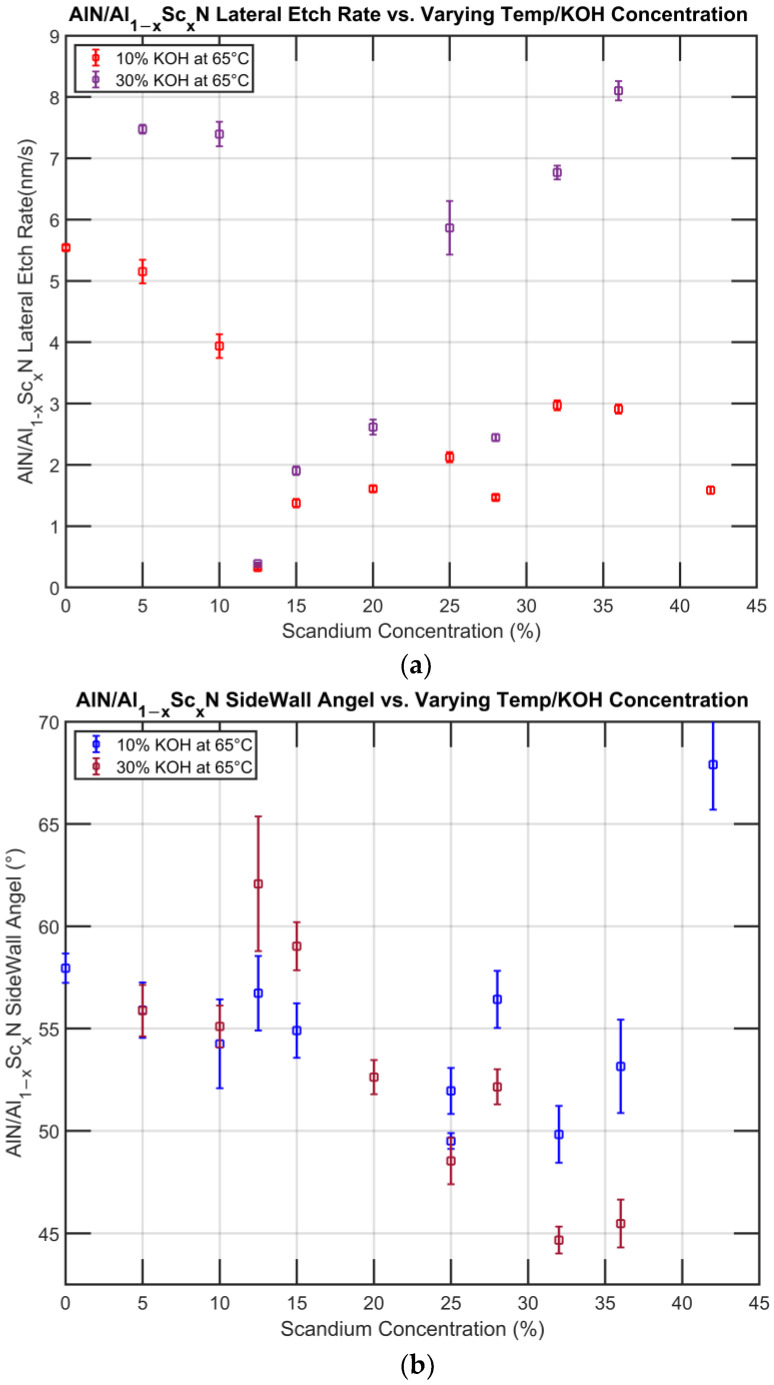
(**a**) Lateral etch rate at elevated temperature and lower etchant concentration. (**b**) Sidewall angle at elevated temperature and lower etchant concentration.

**Figure 12 micromachines-13-01066-f012:**
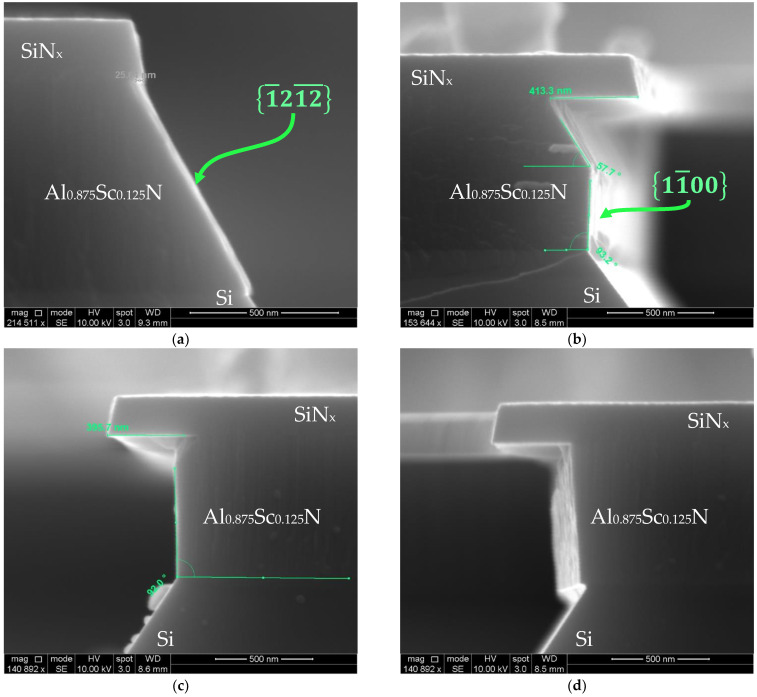
Lateral etch of Al_0.875_Sc_0.125_N in (**a**) 30% KOH at 45 °C for 10 min; (**b**–**d**) 10% KOH at 65 °C for 20 min.

**Figure 13 micromachines-13-01066-f013:**
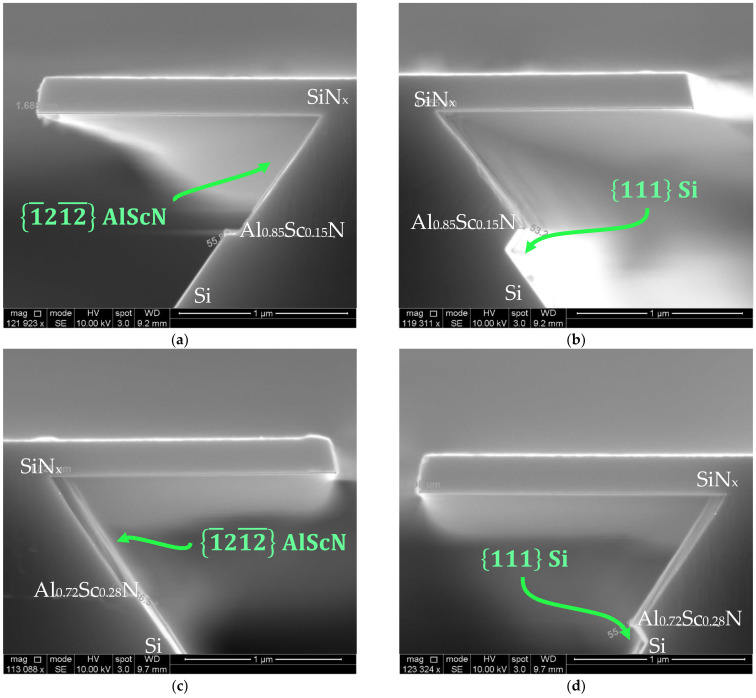
(**a**) Al_0.85_Sc_0.15_N etched for 20 min in 10 wt% KOH, etch front of AlScN approaching Si $\left\{111\right\}$; (**b**) KOH preferentially etches Si $\left\{111\right\}\;$ instead of $\left\{\bar{1}2\bar{12}\right\}$ AlScN; (**c**) Al_0.72_Sc_0.28_N etched for 20 min in 10 wt% KOH, etch front of AlScN aligned with Si $\left\{111\right\}$; (**d**) KOH preferential etching of Si $\left\{111\right\}$ instead of $\left\{\bar{1}2\bar{12}\right\}$ AlScN. All etches were performed at 60 °C.

**Table 1 micromachines-13-01066-t001:** Published dry etch rates of AlN/Al_1−x_Sc_x_N in ICP-RIE.

Material	Crystallinity	Etchant Flow (sccm)	ICP/RF Power (W)	Pressure (m Torr)	Etch Rate (nm·s^−1^)	Rate to AlN	Ref.
AlN	Polycrystal	BCl_3_/Cl_2_/Ar (10/14/6)	800/NA	5	420	1.00	[12]
Al_0.85_Sc_0.15_N		BCl_3_/Cl_2_/He (30/90/100)	550/150	NA	160	0.64	[13]
Al_0.73_Sc_0.27_N		SiCl_4_ (NA)	150/225	15	10	0.42	[14]
Al_0.64_Sc_0.36_N		BCl_3_/Cl_2_/Ar (30/90/70)	400/120	NA	30	0.10	[15]
AlN	Single Crystal	BCl_3_/Cl_2_/Ar (10/20/10)	200/50	5	86.0	1.00	[16]
Al_0.98_Sc_0.02_N					13.3	0.15	
Al_0.84_Sc_0.16_N					11.0	0.127	

**Table 2 micromachines-13-01066-t002:** Published vertical etch rates of AlN/Al_1−x_Sc_x_N in aqueous KOH solutions.

Material	Crystallinity	Etchant	Temp (°C)	Etch Rate (nm·s^−1^)	Activation Energy (kcal·mol^−1^)	Rate at 45 °C (nm·s^−1^)	Ref.
AlN	Single Crystal	AZ400K	60	1.1	15.13	0.8	[26]
	Single Crystal (with high defect density)	AZ400K	60	25.1	15.24	10.3	
	Polycrystal	AZ400K	32	185.0	15.65	561.40	[26]
		AZ400K	40	92.20	2.0 ± 0.5	107.93	[27]
		45 wt% KOH	60	5.83	NA	1.89	[28]
		1 wt% KOH	70	8.33	NA	1.34	[29]
Al_0.80_Sc_0.20_N	Polycrystal	25 wt% KOH	RT	0.66	15.85	3.59	[25]
		25 wt% KOH	40	2.38			
Al_0.64_Sc_0.36_N		25 wt% KOH	80	33.33	NA	2.77	[23]
Al_0.63_Sc_0.37_N		20 wt% KOH	20	0.42	NA	3.56	[30]

Note: For comparison, the etch rates were converted to 45 °C with activation energy *E_a_* given by the author, or assuming *E_a_* = 15.85 kcal/mol if data were unavailable.

**Table 3 micromachines-13-01066-t003:** Scandium concentration vs. Sc target power.

Sc Alloying in Film (%)	0	5	10	12.5	15	20	25	28	30	32	34	36	38	40	42
Sc Target Power (W)	0	40	80	130	185	300	400	450	510	555	610	655	685	710	770

**Table 4 micromachines-13-01066-t004:** Scandium concentration vs. etch time (short etch time in 30% KOH at 45 °C).

Sc Alloying (%)	0	5	10	15	20	25	28	32	36
Etch Time (s)	5	10	20	20	60	60	60	60	60

**Table 5 micromachines-13-01066-t005:** Scandium concentration vs. etch time (long etch in 30% KOH at 45 °C).

Sc Alloying in Film (%)	0	5	10	12.5	15	20	25	28	30	32	34	36	38	40	42
Etch Time (min)	10	10	10	10	5	5	5	5	4	5	4	5	4	4	4

## Data Availability

The data supporting the findings of this study are available from the corresponding author upon reasonable request.

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
