# Peer review of "Vertical and Lateral Etch Survey of Ferroelectric AlN/Al1−xScxN in Aqueous KOH Solutions"

_micromachines, 2022, doi:10.3390/mi13071066_

Round 1

Reviewer 1 Report

Article: Vertical and Lateral Etch Survey of Ferroelectric AlN/Al1-xScxN in Aqueous KOH Solutions. 

This work investigates the vertical and lateral etch rates of sputtered AlN and Al1-xScxN with Sc concentration x ranging from 0 to 0.42 in aqueous potassium hydroxide (KOH). Etch rates and the sidewall angles were reported at different temperatures and KOH concentrations. It was found that the trends of the etch rate were unanimous: while the vertical etch rate decreases with increasing Sc alloying, the lateral etch rate exhibits a V shape transition with a minimum etch rate at x=0.125. By performing an etch on an 800 nm thick Al0.875Sc0.125N film with 10 wt% KOH at 45°C for 10mins, a vertical sidewall was formed by exploiting the ratio.

The manuscript requires significant revisions to improve the manuscript as per the following reviewer’s comments.

1. In Table 1. Use proper text reference in the last column.

2. State the novelty and at the end of the introduction.

3. State the significant contributions and section/subsection details at the end of the introduction.

4. Fig. 3 lacks clarity. Improve the resolution.

5. Follow the proper method of in-text reference-citing as per the journal standards

6. Several abbreviations are not defined at the first appearance in the text.

7. Provide a nomenclature for Abbreviations. Define the abbreviation only at the first appearance.

8. Use proper suffixes for the chemical formula in the text and the references.

9. The conclusion is weak. Provide major conclusions in bulletin form.

10. Improve Grammar and English. 

11. Include further scope of the work.

Reviewer 2 Report

The manuscript titled "Vertical and Lateral Etch Survey of Ferroelectric AlN/Al1-xScxN in Aqueous KOH Solutions," investigates the wet etching anisotropic properties of AlScN at various concentrations of Sc and varying properties of KOH.  Overall with the high demand for AlScN piezoelectric films the ability to pattern the material and alter the lateral and vertical etching profile is of significant importance and is of interest to the readers of Micromachines.  I have some suggestions to improve the manuscript.

1. Try to avoid 1st person terms like We and I.  For instance in the abstract last sentence you say we demonstrated just say it was demonstrated.

2.. In introduction authors have 76o where it looks like they used superscript o instead of degree symbol.  This is also true in XRD results.

3. Deposition parameters, what duty cycle was used for the Pulsed DC, and what was the target distance of the Sc and Al targets?  Was the AlN/ScAlN deposited directly on Si or was there a metal layer?  Was the substrate heated or any RF bias? And how was wafer cleaned.  Need to provide more details on deposition process.

4. Make sure tables and figures have the same format. For instance Table 4 has a different format than other tables. 

5. Figure 9 and 10 caption font is different.

7. Check font and format throughout the manuscript. In some occasions the when describing various concentrations of AlScN the fonts are different like page 15 last paragraph before conclusion.

Reviewer 3 Report

1. Abstract section doesn't provide information about the problem selected. The abstract section must be in standard form.

2. Organization of the manuscript should be checked such as Literature Review and Problem Formulation should be separate sections.

3. References can be limited as per the requirement of the journal.

4. literature must be compared in tabulated form.

5. Conclusion also required presenting in a more quantitative manner.

Round 2

Reviewer 1 Report

The reviewer comments are addressed appropriately.